# Toxic Metals and Metalloids in Infant Formulas Marketed in Brazil, and Child Health Risks According to the Target Hazard Quotients and Target Cancer Risk

**DOI:** 10.3390/ijerph191811178

**Published:** 2022-09-06

**Authors:** Cristine Couto de Almeida, Diego dos Santos Baião, Paloma de Almeida Rodrigues, Tatiana Dillenburg Saint’Pierre, Rachel Ann Hauser-Davis, Katia Christina Leandro, Vania Margaret Flosi Paschoalin, Marion Pereira da Costa, Carlos Adam Conte-Junior

**Affiliations:** 1Graduate Program in Sanitary Surveillance (PPGVS), National Institute of Health Quality Control (INCQS), Oswaldo Cruz Foundation (FIOCRUZ), Rio de Janeiro 21040-900, Brazil; 2Center for Food Analysis (NAL), Technological Development Support Laboratory (LADETEC), Federal University of Rio de Janeiro (UFRJ), Cidade Universitária, Rio de Janeiro 21941-909, Brazil; 3Graduate Program in Veterinary Hygiene (PPGHV), Faculty of Veterinary Medicine, Fluminense Federal University (UFF), Vital Brazil Filho, Niterói 24230-340, Brazil; 4Laboratory of Advanced Analysis in Biochemistry and Molecular Biology (LAABBM), Department of Biochemistry, Federal University of Rio de Janeiro (UFRJ), Cidade Universitária, Rio de Janeiro 21941-909, Brazil; 5Graduate Studies in Food Science (PPGCAL), Institute of Chemistry (IQ), Federal University of Rio de Janeiro (UFRJ), Cidade Universitária, Rio de Janeiro 21941-909, Brazil; 6Department of Chemistry, Pontifical Catholic University of Rio de Janeiro (PUC-Rio), Rua Marquês de São Vicente, 225, Rio de Janeiro 22541-041, Brazil; 7Laboratory for Environmental Health Assessment and Promotion, Oswaldo Cruz Foundation (FIOCRUZ), Rio de Janeiro 21040-900, Brazil; 8Graduate Studies in Chemistry (PGQu), Institute of Chemistry (IQ), Federal University of Rio de Janeiro (UFRJ), Cidade Universitária, Rio de Janeiro 21941-909, Brazil; 9Laboratory of Inspection and Technology of Milk and Derivatives (LaITLácteos), School of Veterinary Medicine and Animal Science, Federal University of Bahia (UFBA), Salvador 40170-110, Brazil

**Keywords:** food analysis, food safety, early childhood health, toxic metals, hazard quotient risk, toxicological risk, carcinogenic risk

## Abstract

Children are highly vulnerable to chemical exposure. Thus, metal and metalloid in infant formulas are a concern, although studies in this regard are still relatively scarce. Thus, the presence of aluminum, arsenic, cadmium, tin, mercury, lead, and uranium was investigated in infant formulas marketed in Brazil by inductively coupled plasma mass spectrometry, and the Target Hazard Quotients (THQ) and Target Cancer Risk (TCR) were calculated in to assess the potential risk of toxicity for children who consume these products continuously. Aluminum ranging from 0.432 ± 0.049 to 1.241 ± 0.113 mg·kg^−1^, arsenic from 0.012 ± 0.009 to 0.034 ± 0.006 mg·kg^−1^, and tin from 0.007 ± 0.003 to 0.095 ± 0.024 mg·kg^−1^ were the major elements, while cadmium and uranium were present at the lowest concentrations. According to the THQ, arsenic contents in infant formulas showed a THQ > 1, indicating potential health risk concerns for newborns or children. Minimal carcinogenic risks were observed for the elements considered carcinogenic. Metabolic and nutritional interactions are also discussed. This study indicates the need to improve infant formula surveillance concerning contamination by potentially toxic and carcinogenic elements.

## 1. Introduction

Adequate and safe nutrition is essential to ensure optimal health, growth, and development during early childhood. Breast milk combines ideal nutritional characteristics with optimal nutrient balance available at all times, at the ideal temperature, and free of microbial contaminants, also favoring numerous immunological and psychological advantages, paramount in reducing infant morbidity and mortality [1,2]. Numerous studies have demonstrated that breastfed infants display significantly decreased risks for asthma, allergy, arthritis, infection, diabetes, cardiovascular disease, obesity, and various cancers in both childhood and adulthood. Not being breastfed, on the other hand, results in loss of immunological protection afforded by maternal colostrum, the “pre-milk” fluid secreted during the first days after delivery, as well as numerous other bioactive compounds that aid in protecting infants in the first two years of life, when the immune and nervous systems are not completely developed [3].

When breastfeeding is impossible or not recommended, infant formulas comprise the most suitable source of nutrients for children under 1 year old, replacing or partially complementing breastfeeding [4]. Infant formulas are industrialized products prepared using modified milk from cows or other animals or hydrolysate proteins from vegetables, such as soy or rice. Cow milk-derived infant formulas are those that most closely resemble the nutritional and functional characteristics of human milk [5]. In Brazil, infant formulas are sold mainly in the powdered form packed in aluminum cans, in contrast to the USA and European countries, where these products are commercialized in both powder and liquid form, ready for consumption, as well as in concentrated liquid form for subsequent dilution [6,7,8].

Environmental contaminants present in infant formulas are a concern, as both the raw materials and/or their manipulation can be subject to inadequate agriculture practices and/or processing methods. In this regard, milk and its derivatives may be contaminated during agricultural management by harmful chemical compounds, such as toxic metals and metalloids, which may be taken up by both plants and animals. Furthermore, industrial processing product packaging also comprise toxic metals and metalloids sources in infant formulas [9].

Metal toxicity in humans depends on several factors, such as their physicochemical properties, ingested frequency and amounts, and individual health *status*, as well as synergistic or antagonistic effects due to the presence of other chemical compounds [9]. These compounds enter the body mainly via the oral route and may affect the central and peripheral nervous systems and gastrointestinal tract, impairing hematopoiesis, and the renal and cardiovascular systems [10]. In this regard, newborns and young children are more vulnerable and sensitive to the harmful effects of toxic metals and metalloids compared to adults, due to inherent rapid childhood growth, physiological organ immaturity and greater susceptibility of the central nervous system (CNS) during the first year of life [9].

The high degree of toxicity of some elements, such as mercury (Hg), cadmium (Cd), arsenic (As), lead (Pb), uranium (U), tin (Sn), and aluminum (Al), makes them of great importance for public health, mainly due to its ability to bioaccumulate in the body [11,12]. Due to this, concerns regarding the quality of milk and dairy products are noted, as these comprise the main food sources during early childhood, and maximum acceptable threshold for several contaminants in milk and dairy products have been established [13]. Consequently, infant formula manufacturers must strictly comply with the established standards to ensure the safety of these products [12].

In this context, the purpose of the present work was to determine the level of toxic elements in infant formula brands recommended for children from 0 to 6 months of age (starting formulas, phase 1) and for children aged 6 to 12 months (follow-up, phase 2) marketed in Brazil and identify the potential health risk from detected levels to both age classes who consume these products continuously.

## 2. Material and Methods

### 2.1. Sample Selection

Three batches of each infant formula widely marketed in Brazil were purchased from different commercial establishments located in the metropolitan region of the city of Rio de Janeiro. All these infant formulas are marketed in a great number of commercial establishments, located at the most populated region of Rio de Janeiro municipality.

All products are registered at the Brazilian regulatory agency (ANVISA) and were packaged in labeled cans in powdered form. Regarding composition, the following inclusion criteria were used for infant formula selection: all formulas should (1) be composed essentially of cow milk; (2) contain unhydrolyzed cow milk proteins; (3) contain lactose as the main carbohydrate; and (4) be supplemented with essential long chain fatty acids as docosahexaenoic acid (DHA) and arachidonic acid (ARA) for children aged from 0 to 6 months (phase 1 starting formulas) and from 6 to 12 months (phase 2 follow-up formulas).

Infant formulas containing protein sources other than cow milk, such as soy or wheat proteins, or developed for specific needs, such as lactose-free or hydrolyzed, were not included in this study. Considering products sold in the Brazilian market that meet the inclusion criteria, ten brand formulas from three different manufacturers were selected. Therefore, five phase 1 and five phase 2 formulas comprising three distinct batches of each brand were included, totaling thirty samples (N = 30). Due to ethical reasons, manufacturers and brands are not disclosed, and samples were coded for the analyses. Brands are represented by two capital letters, followed by 1 or 2 representing phase 1 or phase 2 formulas and the three batches of each brand were identified using capital letters (A, B, or C) (Appendix A).

### 2.2. Reagents

All reagents were of analytical grade. Ultra-pure water (resistivity > 18.2 MΩ cm) was obtained from a Milli-Q^®^ INTEGRAL 10 system (Millipore Co., Billerica, MA, USA). Nitric acid (Vetec, RJ, BRA) was purified by sub-boiling followed by distillation in a quartz still (Kürner Analysentechnik, ROS, DEU). A multi-element standard solution Merck IV (Merck, SP, BRA), comprising 29 elements diluted in nitric acid, was employed to construct the calibration curves.

### 2.3. Sample Preparation and Metal and Metalloid Determinations

Metals and metalloids were determined according to the US EPA method 6020B [14] by inductively coupled plasma mass spectrometry (ICP-MS), with some modifications, as performed in previous studies by the same authors [15]. Briefly, 100 mg of each sample were mixed with 1 mL of concentrated sub-boiled distilled nitric acid and left overnight at room temperature. The following day, the samples were heated at 100 °C for about 4 h in the closed vessels to avoid the volatilization of certain elements, such as Hg. After cooling, samples were diluted with 10 mL of ultra-pure water and analyzed employing a NexIon 300X ICP-MS (PerkinElmer, CT, USA). Instrumental ICP-MS conditions are shown in Table 1. Procedural blanks and two certified reference materials (Skimmed Milk Powder ERM^®^ BD150 and Non-Fat Milk Powder 1549) were also analyzed, in triplicate, to ensure method accuracy.

The total metal and metalloid contents in the infant formula samples were expressed as mg·kg^−1^. The limit of detection (LOD) and limit of quantification (LOQ) were determined according to the Brazilian National Institute of Metrology, Quality, and Technology—INMETRO [16], where LOD were expressed as 3 × SD/Slope of the curve and LOQ expressed as 10 × SD blank/Slope of the curve.

### 2.4. Infant Potential Health Risk

#### 2.4.1. Target Hazard Quotient

The target hazard quotient (THQ), which measures the toxicological risk of certain contaminants, non-carcinogenic ones, over a specific population [17], was calculated to assess child exposure to the toxic metals and metalloids identified in infant formulas. The THQ was calculated using Equation (1), described below. The ratio between the ingested contaminant was calculated from the recommended infant formula ingestion rate (following labeling recommendations) and the sub-chronic oral reference dose (RfDo) for each toxic element established by the United States Environmental Protection Agency [18] and the Agency for Toxic Substances and Disease Registry [19] for the general population. A THQ ≤ 1, suggests minimal evident risk, whereas a THQ > 1 indicates a potential harmful risk of adverse effects on infant health.
(1)$$\mathrm{THQ}=(\mathrm{EF}\;\times \;\mathrm{ED}\;\times \;\mathrm{IFIR}\;\times \;C_{\mathrm{element}}/\mathrm{RfDo}\;\times \;\mathrm{WAB}\;\times \;\mathrm{TA})\times 10^{-3}$$
where EF comprises the exposure frequency of 182.5 days, ED is the exposure duration, corresponding to the recommended consumption period for each type of infant formula, where 0.5 = 6 months of product consumption, IFIR consists in the daily infant formula ingestion (Appendix A), C_element_ is the concentration of each toxic metal or metalloid in the infant formula (mg·kg^−1^), RfDo comprises an estimate of the safe daily oral exposure, expressed as units of mg·kg^−1^∙day^−1^ (Appendix A), WAB is the average body weight reference, in this case referring to ages from 0 to 6 months old (6.1 kg) and from 7 to 12 months old (9.1 kg) [1], and TA is the average exposure time for human non-carcinogens (EF × ED) and 10^−3^ is the unit conversion factor (performed through a dimensional analysis).

If the THQ value for non-carcinogenic metals or metalloids is lower than 1, an unlikely health risk is suggested, while values over 1 indicate a likely health risk, although non-carcinogenic, with an increasing probability with increasing target hazard quotients.

#### 2.4.2. Target Cancer Risk

The target cancer risk (TCR), which is used to assess the potential risk associated with exposure to carcinogens during a lifetime exposure period, was calculated for Al, inorganic As (iAs), and Pb, as these elements present potential carcinogenic effects [17,20]. Instead of the RfDo, the TCR was determined by using the slope factor (CSF_O_) stablished by each element to estimate the probability of an individual developing cancer following oral ingestion of contaminant elements considering the body mass, determined with the equation below:(2)$$\mathrm{TCR}=(\mathrm{EF}\;\times \;\mathrm{ED}\;\times \;\mathrm{IFIR}\;\times \;C_{\mathrm{element}}\;\times \;\mathrm{CSFo}/\mathrm{WAB}\;\times \;\mathrm{TA})\times 10^{-3}$$
where EF is the exposure frequency (182.5 days), ED is the exposure duration corresponding to the recommended consumption period for each type of infant formula (0.5 = 6 months of product consumption), IFIR consists in the daily infant formula ingestion (Appendix A), C_element_ is the metal concentration in the infant formula (mg·kg^−1^), CSF_O_ is the oral cancer slope factor for Al (0.021 mg·kg^−1^), iAs (1.5 mg·kg^−1^), and Pb (0.0085 mg·kg^−1^) [18,21], WAB is the average body weight reference, in this case referring to ages from 0 to 6 months old (6.1 kg) and from 7 to 12 months old (9.1 kg) [1], and TA is the averaged exposure time to the carcinogen (EF × ED) and 10^−3^ is the unit conversion factor. Oral cancer slope factors were established only for elements that are considered carcinogenic.

### 2.5. Statistical Analyses

Significant differences in element contents among the infant formula batches were assessed through a two-way analysis of variance (ANOVA) with repeated measures. A post hoc analysis (Bonferroni correction) was performed when a significant *F* was found. The results were considered statistically significant when *p* < 0.05 and were expressed as the means ± standard deviations (SD). All statistical analyses were carried out using the GraphPad Prism^®^ (GraphPad Software, San Diego, CA, USA).

## 3. Results and Discussion

### 3.1. Method Accuracy and Precision

The observed and certified values for the reference materials Skimmed Milk Powder ERM^®^ BD150 and Non-Fat Milk Powder 1549 CRMs are displayed in Table 2. Recoveries ranged from 87.1% to 120%, adequate for this type of analysis [22,23]. The LOD and LOQ obtained for each element are displayed in Appendix A.

### 3.2. Metal and Metalloid Contents in Infant Formulas

The mean metal and metalloid concentrations in the three batches of the same brand of phase 1 or phase 2 infant formulas were not significantly different for Cd, Hg, and Pb (Table 3). Furthermore, Hg were not detected in any infant formulas, and Pb was not detected in some phase 1 and phase 2 infant formulas (except for samples NN1, DM1, ME2, and NC2). Concerning the other investigated elements, Al ranged from 0.432 ± 0.049 to 1.241 ± 0.113 mg·kg^−1^, As from 0.012 ± 0.009 to 0.034 ± 0.006 mg·kg^−1^, Sn from 0.007 ± 0.003 to 0.095 ± 0.024 mg·kg^−1^, and U from 0.002 ± 0.001 to 0.016 ± 0.007 mg·kg^−1^, displaying significant heterogeneity between the investigated formulations (*p* < 0.01) (Table 3).

Metals and metalloids are present in all environmental compartments and, thus, present in virtually all foodstuff [24]. Metal uptake by neonates depends on bioavailability from milk diets and may irreversibly affect physical and cognitive developments [9]. Among the toxic elements detected in infant formulas, Al was present at the highest concentrations, followed by Sn and As. The difference observed between Al values in phase 1 and phase 2 infant formulas suggests contamination during manufacturing or storage, as Al containers and tin packs lined with Al foil have been reported as the source of this contamination [25].

The lowest concentrations were observed for Cd and U, whereas Hg was not detected, and Pb was detected in only three sample (samples NN1, 0.036 ± 0.041 mg·kg^−1^; DM1, 0.016 ± 0.021 mg·kg^−1^, ME2, 0.023 ± 0.033 mg·kg^−1^, and NC2, 0.011 ± 0.001 mg·kg^−1^). The toxic elements reported herein are similar to those reported in twenty-six infant formulas marketed in Nigeria, where Al ranged from 0.41 to 2.47 mg·kg^−1^ and As ranged from 0.02 to 1.56 mg·kg^−1^. Hg, Sn, U, and Pb contents, however, were not assessed [26]. In another assessment, thirty infant formulas from 15 different brands marketed on the island of Tenerife (Spain) were evaluated, with Al contents ranging from 1.58 to 7.05 mg·kg^−1^, Cd from 0.01 to 0.002 mg·kg^−1^, and Pb from 0.05 to 0.09 mg·kg^−1^ [27], with Sn, As, U, and Hg not being evaluated. In Sweden, nine formulas presented high As levels, ranging from 0.17 to 1.58 µg·L^−1^ and Pb, ranging from 0.82 to 1.50 µg·L^−1^, with Sn, Hg, and Al not assessed [28].

The Codex Alimentarius indicates that these contaminants are not intentionally added to foodstuff, and may contaminate food items during food processing and/or storage or due to environmental contamination in raw materials [13]. Milk and by-product contamination can occur during various manufacturing stages, either intentionally, following food additive addition for technological purposes, or unintentionally through environmental contamination (ex., soil and grassland contamination), raw material contamination (i.e., cattle consumption of contaminated food and water), or transfer from the packaging to the final product [9]. In addition, the drinking water used by consumers in the reconstitution of powdered infant formulas can also significantly increase toxic elements concentrations when compared to ready-to-eat products [28,29], and infants fed by formulas reconstitute with tap water are at the highest risk for metals and metalloids contamination from contaminated water supplies [24]. In view of this, tolerable and safe infant metal and metalloid contents have been established by the Agency for Toxic Substances and Disease Registry (2005 and 2008) and United States Environmental Protection Agency (U.S. EPA, 2022) (Appendix A).

The great variability of Al content found in infant formulas suggests that contamination occurs during manufacturing or storage in those containers Al cans, in glass containers, or tin pack lining with Al foil [25]. In addition to Al, the presence of other chemical substances such as As, Cd, Sn, and U in milk can occur due to environmental contamination in soil or grasslands, where indiscriminate use of chemicals is used in cattle management and milk production or can either occur during milk storage and transportation to industrial plants [9].

### 3.3. Metal and Metalloid Contents in Different Infant Formula Batches from the Same Manufacturer

No inter-batch and brand differences were observed for As and Cd contents (*p* < 0.01). Furthermore, in the majority of phase 1 and phase 2 infant formulas, the Hg was not detected, and Pb was detected in only three sample, NC1C, DM1B, and NC2A (Table 4). Concerning Al, inter-batch variations were observed in samples ME1, NN1, and ME2. Similar behavior was noted for Sn, with inter-batch differences observed for samples NC1, NN1, DA1, NC2, and NN2 samples. Inter-batch variations were observed for U concerning samples ME1, DM2, and DA2. Significant differences are displayed in Table 4. Regarding Al, the NC1C sample showed 1.008 ± 0.178 mg·kg^−1^; DA1C, 1.031 ± 0.096 mg·kg^−1^; NC2B, 1.142 ± 0.221 mg·kg^−1^ and the three lots of the DA2 brand showed, respectively, 1.294 ± 0.146, 1.111 ± 0.129 and 1.316 ± 0.132 mg·kg^−1^, the higher concentrations. Samples NN1C, 0.481 ± 0.054 mg·kg^−1^; DM1B, 0.441 ± 0.097 mg·kg^−1^; DM1C, 0.476 ± 0.158 mg·kg^−1^; and DM2A, 0.434 ± 0.058 mg·kg^−1^ showed the lower contents when compared with other brands. Regarding Sn, NC1B samples showed a content of 0.105 ± 0.035 mg·kg^−1^; NN1B, 0.101 ± 0.025 mg·kg^−1^; DM1B, 0.093 ± 0.026 mg·kg^−1^; NC2B, 0.120 ± 0.015 mg·kg^−1^; NC2C, 0.093 ± 0.024 mg·kg^−1^; and NN2B, 0.098 ± 0.033 mg·kg^−1^, all of them, superior to the concentrations found in ME1A, 0.007 ± 0.003 mg·kg^−1^; ME1B, 0.013 ± 0.004 mg·kg^−1^; NN1C, 0.028 ± 0.006 mg·kg^−1^; ME2C, 0.023 ± 0.014 mg·kg^−1^; and DM2C, 0.024 ± 0.005 mg·kg^−1^.

Raising health risk awareness concerning metal and metalloid contamination is important for risk management during infancy, particularly for neonates, as this group is more susceptible to toxic effects due to considerable variations in elemental absorption, distribution, metabolism, and excretion [30]. In newborns, the gastric pH ranges from 6 to 8, and metal and metalloid bioavailability increases at alkaline pH. Furthermore, toxic metal and metalloid effects may also occur due to deficient enzyme systems and ionic imbalances between essential and non-essential metals [31]. Rapid infant growth and development in the first year of life coupled with high energy requirements and high food consumption relative to body weight make infants more vulnerable to this type of contamination compared to adults [9].

In addition, non-essential metals and metalloids may result in potential toxic effects even at extremely low concentrations. In this regard, Hg, Cd, Pb, and Al can interact with essential minerals, such as Ca, Cu, Zn, Fe, Mn, and Se in the human body [32,33]. Due to shared common chemical characteristics, these elements may compete for ligands in biological systems, decreasing the gastrointestinal absorption of certain minerals and substituting metallic ions in several cellular processes, compromising enzymatic effectiveness and endurance [11,34,35]. Furthermore, Al can also be deposited in bone, brain, heart, spleen, and muscles, resulting in cumulative effects with exposure times. Another factor that must be considered when assessing dietary infant exposure to toxic metals and metalloids is the rapid growth and development during the first year of life, as well as energy requirements and, thus, food consumption, which is much higher relative to intact body weight when compared to elderly children and adults [34].

Correlating the findings in this study with those found in previous works, in which the objective was to evaluate the contents of essential minerals in the same infant formula samples and with the same analysis methodology, we observed that, in general, the concentrations found for essential minerals were significantly higher than those observed for toxic elements [15]. However, it is worth noting that it is not possible to directly correlate the metabolic interference of non-essential metals with essential minerals since the analysis was performed directly on the product and not after its ingestion.

### 3.4. Estimating Infant Health Risk for Toxic Metals and Metalloids Found in Infant Formulas

The recommended method for assessing cumulative risk and hazard to human beings following exposure to multiple chemicals is based on the default approaches described in the Guidelines for the Health Risk Assessment of Chemical Mixtures, published in 1986 [17]. Additional improvement on this methodology was subsequently published in the Supplementary Guidance for Conducting Health Risk Assessment of Chemical Mixtures in 2000b, but both have recommended using the target hazard quotient and target risk cancer to evaluate the safety of foodstuff containing toxic metals and/or metalloids use with quantitative cancer risk estimates as well as with HQs should be used [17]. The oral cancer slope factors (CFSo) are considered to estimate cancer risk, and the reference oral doses (RfDo) are used to calculate the hazard quotient. Both values refer to the general population, including sensitive subgroups. In this study, in order to consider the age group evaluated, the average body weights for each age group (0–6 months and 7–12 months) were considered for the risk calculations [17,18,34]. Dose–response assessments regarding carcinogen risk define the relationship between the dose of a toxic agent and its putative carcinogenic effect if it is classified as a carcinogenic for human beings. Data obtained in dose–response assessments are combined with exposure assessment data to estimate risks. When the risks to short exposure or prolonged exposures cannot be measured using animal models or data from epidemiologic studies, mathematical models should be applied. When data are limited and when the mechanisms of carcinogenic action are uncertain, linear models or procedures are adopted if compatible with the available information. The U.S. Environmental Protection Agency (U.S. EPA) usually employs the linearized multistage procedure in the absence of adequate information [17,20].

Furthermore, the values calculated for the THQ and TCR indexes were based on the average of the values analyzed for each toxic metal or metalloid in infant formulas. Therefore, the values used to calculate these estimates did not address the best or worst scenario, but rather the average of toxic metals determined in infant formula.

#### 3.4.1. Target Hazard Quotient

To assess the potential toxic risk to a particular contaminant, the THQ was calculated for Hg, Cd, Pb, Sn, Al, U, and As. Values THQ < 1.0 indicate minimal evident risk, and THQ > 1.0 indicates potential harmful effects [17]. The THQ calculated for infants considering each specific element concentration were calculated for each of the three batches of phase 1 and phase 2 infant formulas (Table 5). Considering the average consumption of phase 1 and phase 2 infant formulas, minimal toxicity risks were observed for Hg, Cd, Pb, Sn, Al, and U (THQ ≤ 1.0). On the other hand, significant potential risks regarding As were observed in the most phase 1 and phase 2 infant formulas.

Harmful outcomes aside from cancer and genetic mutations due to chemical exposure are termed systemic toxicities, due to their general potential physio-pathological effects on various organs in the human body. Based on our understanding of homeostasis and adaptive mechanisms, systemic toxicants should be considered if there is an identifiable exposure threshold for each individual and for populations below which there are no observable adverse effects [36]. Traditionally, the systemic effects have been evaluated from terms such as acceptable daily intake (ADI), safety factor (SF), and margin of safety (MOS), taking into account a target limit. In 1983, the U.S. EPA formed a committee called RfD Work Group to carry forward this issue, which prepared a risk assessment report to establish regulatory decisions for the use and/or exposure to non-cancer, non-mutagenic compounds, based on experimental data. The RfD data should be considered reference points to gauge the potential effects of chemical compounds at different doses. Usually, doses inferior to RfD are not likely to be associated with adverse health risks and are exempt of regulatory compliance. When the frequency and/or magnitude of the exposure exceed the RfD, the possibility of adverse effects in humans increases; however, it should not be categorically concluded that all doses below the RfD are “acceptable” (or would be risk-free) and that all doses over the RfD are “unacceptable” (or would result in adverse effects) [36].

In the present study, the estimation of a daily intake of toxic elements found in infant formulas was compared to RfD, which reflects the threat of ingesting toxic elements through infant formulas consumption, and the THQs were calculated to evaluate the risk of adverse health effects to a certain toxic element [17]. The THQ values estimated for all these toxic elements were <1, except for inorganic As, which pointed to a potential risk following the ingestion of almost all phase 1 and phase 2 infant formulas. Therefore, the Hg, Cd, Pb, Sn, Al, and U contents determined herein indicate that the estimated child exposure to these toxic metals following infant formulas consumption does not represent risks to young child health in Brazil. However, considering THQ for iAs exposure following intake of infant formulas, the child health risk increased. Furthermore, when comparing batches, THQ differences were observed between phase 1 and 2 infant formulas, explained by the fact that several elements varied significantly between brands, resulting in variations in the daily intakes. It is important to note that food diversification is introduced to infants from the sixth month onwards. Considering the possibility of the presence of metals and metalloids in other foods and possible bioaccumulation processes, the calculated THQs for phase 2 infant formulas should be taken in account in addition to other likely sources of iAs contamination.

As is a metalloid element sharing metal and nonmetal features and widely distributed in the biosphere in inorganic or organic forms. As can easily enter the food chain through contaminated soil or water because soil is the major sink of As, summed to its high mobility and rate uptake rate. As is released from natural sources (parent rocks), from agricultural practices, irrigation with As-contaminated water, improper application of arsenical fertilizers, insecticides, or herbicides, mining activities, and petroleum refineries [37,38].

As in organic forms and bound to macromolecules are known to be less toxic than iAs species. There is no evidence that As is essential to human cellular processes, but on the other hand, exposure to As has been linked to severe health conditions such as hyperkeratosis, gangrene, hypertension, peripheral vascular disease, melanosis, and keratosis [37,38]. As has no single major mode of action in the human body, but a few well-documented mechanisms describing arsenic enrollment on cancer are related to an increase in hydrogen peroxide and superoxide anions production; interaction with cysteine residues in zinc finger domains of transcriptional factors, deregulating cell proliferation, and inducing epigenetic alterations. Since cancer risk may be non-linear, a safe level cannot be determined by extrapolating risk from high dose exposures [39].

#### 3.4.2. Target Cancer Risk

In order to evaluate the probable cancer risk of As, Al, and Pb, an incremental lifetime cancer risk (ILTCR) is determined by multiplying the exposure to iAs, Al, and Pb by the appropriate cancer slope factor (CSF) [20]. For As, the range bound levels were 1.5 mg·kg·day^−1^ and 4.5 mg·kg·day^−1^ for adults and children, respectively, approaching a 95% confidence level. Furthermore, the CSF estimated for Al and Pb were 0.021 mg·kg·day^−1^ and 0.0085 mg·kg·day^−1^, respectively [18,21]. The TCR values for Al ranges from 0.0001 to 0.0004 (1.0 × 10^−4^ to 4.0 × 10^−4^), As ranges from 0.0002 to 0.0009 (2.0 × 10^−4^ to 9.0 × 10^−4^), and for Pb ranges from 0.000001 to 0.000006 (1.0 × 10^−6^ to 6.0 × 10^−6^) (Table 5).

Younger populations may be particularly at risk for long-term health effects of chronic exposure to iAs from milk or rice-derived based products for infants and young children because of the massive exposure during earlier years of life and while they are still developing different systems and the presumed long-term exposure. As exposure seems to provoke internal cancers in organs such as skin, lung, liver, kidney, and bladder. Therefore, iAs has been classified by the International Agency for Research on Cancer as a non-threshold since even small doses may be associated with cancer risk, being considered a class 1 human carcinogen [37,38]. However, in the present study, negligible carcinogenic risks for infants were raised considering Al, As, and Pb since the estimated levels are within the acceptable risk level for developing cancer established by the World Health Organization at 10^−5^ mg·kg·day^−1^ and by U.S. EPA, ranging from 10^−4^ to 10^−6^ mg·kg·day^−1^ [40].

Studies have shown that breastfed babies have less exposure to As than formula-fed babies and receive better nutrition. Attention should be taken if infant formulas offered to infants have rice-derived based products in their formulation, which are commonly contaminated by abundant species of As, including the most toxic to humans [40]. Additionally, it is recommended to use bottled water to resuspend the infant formula to be sure that the domestic water supply has minimal As concentrations. It should be remembered that the manufacturing conditions, consumption (mixing with water or milk), and the amount consumed may vary according to geography, culture, and individual characteristics and this may affect the exposure to heavy metals. Controlling the baby foods processing at every manufacturing step associated with systematic monitoring of heavy metals and ingestion of infant formulas may provide significant contributions to reducing exposure to carcinogenic elements by dietary intake [9].

## 4. Conclusions

The Brazilian phase 1 and 2 infant formulas contained undetectable Hg values, but low contents of Cd, Pb, and U. However, Al, As, and Sn were found in high contents. The THQ values estimated for Hg, Cd, Pb, Al, Sn, and U were inferior to 1 (THQ < 1), indicating that Brazilian infant formulas do not represent a health risk for early childhood for the aforementioned toxic metals. On the other hand, the THQ calculated for iAs was over 1 (THQ > 1), indicating potential risks regarding As ingestion in most phase 1 and phase 2 infant formulas. However, negligible TCR for infants were noted regarding As, Al, and Pb, as these toxic metals/metalloids were within the safe limits established by international regulatory organizations.

Based on the data reported herein, caution is recommended during the inspection of the investigated formulas by public health authorities regarding contamination by toxic metals and metalloids, especially the excess As consistently identified in most infant formulas. Further attention given to the inspection of infant formulas should guarantee the reliability of these products in relation to international public health organizations recommendations on toxic metals and metalloids, beyond nutritional quality, the protein source used to supply the needs of early childhood, and labeling information.

The present study encourages a complete risk assessment of the entire production chain in order to guide regulatory and sector risk managers on maximum ingestion limits for newborns and infants that should be established for metal or metalloids concerning infant formulas marketed in Brazil.

## Figures and Tables

**Table 1 ijerph-19-11178-t001:** ICP-MS conditions applied for elemental determinations.

ICP-MS Condition	Value
Radiofrequency power	1100 W
Plasma flow	17.0 L·min^−1^
Auxiliary gas flow	1.2 L·min^−1^
Carrier gas flow	0.98 L·min^−1^
Skimmer composition	Pt
Dwell time	50 ms per isotope
Scanning mode	Peak hopping
Resolution	0.7 uma (u)
Scans by reading	5

**Table 2 ijerph-19-11178-t002:** Toxic metals determination in reference materials and recovery values (%) for each mineral relative to the reference concentration value.

Toxic Elements	Reference Materials (mg·kg^−1^)					
	Skimmed Milk Powder BD150^®^			Non-Fat Milk Powder 1549^®^		
	Experimental	Reference	Recovery	Experimental	Reference	Recovery
Hg	0.072 ± 0.035	0.06 ± 0.007	120%	-	-	-
Cd	0.010 ± 0.005	0.0114 ± 0.0029	87.1%	0.0006 ± 0.011	0.0005 ± 0.0002	120%
Pb	0.018 ± 0.870	0.019 ± 0.004	94.7%	0.019 ± 0.016	0.019 ± 0.03	100%

The certified reference materials (CRM) were analyzed by inductively coupled plasma mass spectrometry (ICP-MS). The CRMs certificates did not present reference values for the elements Hg (Milk Powder 1549), and As, Sn, Al, and U (Skimmed Milk Powder ERM^®^ BD150 and Non-Fat Milk Powder 1549).

**Table 3 ijerph-19-11178-t003:** Average concentrations of toxic elements in phase 1 and phase 2 infant formulas marketed in Brazil.

IFs	Toxic Elements (mg·kg^−1^)						
	Al	As	Cd	Sn	Hg	Pb	U
ME1	0.724 ± 0.141 ^b,c^	0.016 ± 0.004 ^b^	0.004 ± 0.001 ^a^	0.007 ± 0.003 ^c^	<LOQ	<LOD	0.005 ± 0.002 ^b,c^
NC1	0.459 ± 0.177 ^c,d^	<LOQ	0.005 ± 0.002 ^a^	0.081 ± 0.022 ^a^	<LOQ	<LOD	<LOD
NN1	0.504 ± 0.099 ^c^	<LOQ	0.004 ± 0.002 ^a^	0.054 ± 0.023 ^a,b^	<LOQ	0.036 ± 0.041	<LOD
DM1	0.432 ± 0.049 ^d^	0.031 ± 0.003 ^a^	0.005 ± 0.002 ^a^	0.068 ± 0.022 ^a,b^	<LOD	0.016 ± 0.021	0.009 ± 0.001 ^a^
DA1	0.746 ± 0.189 ^a,b,c^	0.020 ± 0.009 ^a,b^	0.005 ± 0.002 ^a^	0.040 ± 0.011 ^b^	<LOD	<LOD	0.011 ± 0.001 ^a^
ME2	0.673 ± 0.215 ^b,c,d^	0.021 ± 0.007 ^a,b^	<LOQ	0.010 ± 0.002 ^c^	<LOD	0.023 ± 0.033	0.007 ± 0.003 ^a,b^
NC2	0.942 ± 0.200 ^a,b^	0.012 ± 0.009 ^b^	0.008 ± 0.003 ^a^	0.095 ± 0.024 ^a^	<LOD	0.011 ± 0.001	0.003 ± 0.001 ^c^
NN2	0.494 ± 0.138 ^c,d^	0.015 ± 0.005 ^b^	0.004 ± 0.002 ^a^	0.075 ± 0.021 ^a^	<LOD	<LOD	0.002 ± 0.001 ^c^
DM2	<LOQ	0.034 ± 0.006 ^a^	0.004 ± 0.001 ^a^	0.033 ± 0.011 ^b^	<LOD	<LOD	0.009 ± 0.002 ^a,b^
DA2	1.241 ± 0.113 ^a^	0.024 ± 0.008 ^a,b^	0.004 ± 0.002 ^a^	0.035 ± 0.012 ^b^	<LOD	<LOD	0.016 ± 0.007 ^a^

Data refer to averages from three batches of each brand of infant formulas marketed in Brazil. Samples were analyzed in quintuplicate and data are reported as means ± SD (N°: 150). Different superscript letters in the same column indicate significant differences between the IFs at a significance level of *p* < 0.01. IFs, Infant formulas; LOD, limit of detection; LOQ, limit of quantification. LOD: Hg = 0.003; Pb = 0.003; U = 0.0004. LOQ: Al = 0.430; As = 0.011; Cd = 0.002; Sn = 0.005; Hg = 0.011; Pb = 0.010; U = 0.001. Toxic metals and metalloid concentrations were determined using inductively coupled plasma mass spectrometry (ICP-MS).

**Table 4 ijerph-19-11178-t004:** Average concentrations of toxic metals and metalloids in phase 1 and phase 2 infant formula in distinct batches.

IFs (Batches)	Toxic Elements (mg·kg^−1^)						
	Al	As	Cd	Sn	Hg	Pb	U
Phase 1 infant formulas							
ME1A	0.561 ± 0.048 ^b^	0.013 ± 0.005 ^a^	<LOQ	0.007 ± 0.003 ^a^	<LOQ	<LOD	0.002 ± 0.001 ^b^
ME1B	0.806 ± 0.017 ^a^	0.021 ± 0.008 ^a^	0.002 ± 0.001 ^a^	0.013 ± 0.004 ^a^	<LOQ	<LOD	0.006 ± 0.003 ^a,b^
ME1C	0.804 ± 0.080 ^a^	0.015 ± 0.001 ^a^	0.003 ± 0.001 ^a^	<LOQ	<LOQ	<LOD	0.006 ± 0.001 ^a^
NC1A	<LOQ	<LOQ	0.004 ± 0.001 ^a^	0.076 ± 0.012 ^a,b^	<LOQ	<LOD	<LOD
NC1B	<LOQ	0.014 ± 0.001	0.006 ± 0.004 ^a^	0.105 ± 0.035 ^a^	<LOQ	<LOD	<LOD
NC1C	1.008 ± 0.178	<LOQ	0.004 ± 0.001 ^a^	0.061 ± 0.006 ^b^	<LOQ	<LOD	<LOD
NN1A	<LOQ	<LOQ	0.002 ± 0.001 ^a^	0.033 ± 0.013 ^b^	<LOQ	<LOD	<LOD
NN1B	0.602 ± 0.038 ^a^	<LOQ	0.007 ± 0.004 ^a^	0.101 ± 0.025 ^a^	<LOD	0.036 ± 0.041	<LOD
NN1C	0.481 ± 0.054 ^b^	0.011 ± 0.002	0.003 ± 0.001 ^a^	0.028 ± 0.006 ^b^	<LOQ	<LOD	<LOD
DM1A	<LOQ	0.028 ± 0.004 ^a^	0.003 ± 0.001 ^a^	0.051 ± 0.023 ^a^	<LOD	<LOD	0.009 ± 0.001 ^a^
DM1B	0.441 ± 0.097 ^a^	0.032 ± 0.001 ^a^	0.006 ± 0.003 ^a^	0.093 ± 0.026 ^a^	<LOD	0.016 ± 0.021	0.008 ± 0.001 ^a^
DM1C	0.476 ± 0.158 ^a^	0.033 ± 0.004 ^a^	0.005 ± 0.001 ^a^	0.062 ± 0.016 ^a^	<LOD	<LOD	0.009 ± 0.001 ^a^
DA1A	0.904 ± 0.225 ^a^	0.028 ± 0.004 ^a^	0.005 ± 0.001 ^a^	0.030 ± 0.005 ^b^	<LOD	<LOQ	0.010 ± 0.001 ^a^
DA1B	<LOQ	0.023 ± 0.005 ^a^	0.007 ± 0.003 ^a^	0.039 ± 0.005 ^b^	<LOD	<LOD	0.011 ± 0.001 ^a^
DA1C	1.031 ± 0.096 ^a^	<LOQ	0.003 ± 0.001 ^a^	0.052 ± 0.006 ^a^	<LOD	<LOD	0.012 ± 0.001 ^a^
Phase 2 infant formulas							
ME2A	0.916 ± 0.047 ^a^	0.022 ± 0.001 ^a^	<LOQ	<LOQ	<LOD	<LOQ	0.008 ± 0.001 ^a^
ME2B	0.507 ± 0.021 ^b^	0.025 ± 0.005 ^a^	<LOQ	<LOQ	<LOD	0.023 ± 0.033	0.007 ± 0.001 ^a,b^
ME2C	0.596 ± 0.060 ^b^	0.015 ± 0.006 ^a^	0.002 ± 0.001	0.023 ± 0.014	<LOD	<LOQ	0.005 ± 0.001 ^b^
NC2A	0.742 ± 0.258 ^a^	<LOQ	0.007 ± 0.002 ^a^	0.073 ± 0.011 ^b^	<LOD	0.011 ± 0.001	0.002 ± 0.001 ^a^
NC2B	1.142 ± 0.221 ^a^	0.012 ± 0.001 ^a^	0.010 ± 0.001 ^a^	0.120 ± 0.015 ^a^	<LOD	<LOQ	0.004 ± 0.001 ^a^
NC2C	0.942 ± 0.354 ^a^	0.014 ± 0.003 ^a^	0.008 ± 0.002 ^a^	0.093 ± 0.024 ^a,b^	<LOD	<LOQ	0.003 ± 0.001 ^a^
NN2A	0.570 ± 0.108 ^a^	0.014 ± 0.007 ^a^	0.003 ± 0.001 ^a^	0.059 ± 0.005 ^b^	<LOD	<LOQ	0.002 ± 0.001 ^a^
NN2B	0.578 ± 0.081 ^a^	0.014 ± 0.008 ^a^	0.006 ± 0.002 ^a^	0.098 ± 0.033 ^a^	<LOD	<LOQ	0.002 ± 0.001 ^a^
NN2C	<LOQ	0.018 ± 0.003 ^a^	0.004 ± 0.001 ^a^	0.068 ± 0.014ª^,b^	<LOD	<LOD	0.001 ± 0.001 ^a^
DM2A	0.434 ± 0.058	0.039 ± 0.007 ^a^	0.005 ± 0.002 ^a^	0.046 ± 0.017 ^a^	<LOD	<LOD	0.009 ± 0.001 ^a,b^
DM2B	<LOQ	0.032 ± 0.004 ^a^	0.004 ± 0.001 ^a^	0.030 ± 0.001 ^a^	<LOD	<LOD	0.011 ± 0.001 ^a^
DM2C	<LOQ	0.032 ± 0.002 ^a^	0.004 ± 0.001 ^a^	0.024 ± 0.005 ^a^	<LOD	<LOD	0.008 ± 0.001 ^b^
DA2A	1.294 ± 0.146 ^a^	0.022 ± 0.003 ^a^	0.005 ± 0.001 ^a^	0.037 ± 0.006 ^a^	<LOD	<LOD	0.020 ± 0.002 ^a^
DA2B	1.111 ± 0.129 ^a^	0.026 ± 0.002 ^a^	0.004 ± 0.001 ^a^	0.032 ± 0.004 ^a^	<LOD	<LOD	0.008 ± 0.001 ^b^
DA2C	1.316 ± 0.132 ^a^	0.023 ± 0.003 ^a^	0.004 ± 0.001 ^a^	0.035 ± 0.003 ^a^	<LOD	<LOD	0.020 ± 0.001 ^a^

Samples were analyzed in quintuplicate and data are reported as means ± SD (N°: 150). Different superscript letters in the same column indicate significant differences between lots of each infant formula brands at a significance level of *p* < 0.01. IFs, Infant formulas; LOD, limit of detection; LOQ, limit of quantification. LOD: Hg = 0.003; Pb = 0.003; U = 0.0004. LOQ: Al = 0.430; As = 0.011; Cd = 0.002; Sn = 0.005; Hg = 0.011; Pb = 0.010; U = 0.001.

**Table 5 ijerph-19-11178-t005:** Target hazard quotient (THQ) to the toxic metals and metalloids, and the target cancer risk (TCR) of Al, As, and Pb estimated in batches of phase 1 and phase 2 infant formulas.

IFs	THQ (mg·kg·day^−1^)							TCR (mg·kg·day^−1^)		
	iAs	Hg	Cd	Pb	Al	Sn	U	Al	As	Pb
Phase 1 infant formulas										
ME1A	0.8628	n.d.	n.d.	n.d.	0.0109	0.0002	0.0157	0.0002	0.0004	n.d.
ME1B	1.3339	n.d.	0.0384	n.d.	0.0156	0.0004	0.0398	0.0003	0.0006	n.d.
ME1C	0.9720	n.d.	0.0550	n.d.	0.0156	n.d.	0.0361	0.0003	0.0004	n.d.
NC1A	n.d.	n.d.	0.0803	n.d.	n.d.	0.0026	n.d.	n.d.	n.d.	n.d.
NC1B	0.9361	n.d.	0.1318	n.d.	n.d.	0.0036	n.d.	n.d.	0.0004	n.d.
NC1C	n.d.	n.d.	0.0809	n.d.	0.0206	0.0021	n.d.	0.0004	n.d.	n.d.
NN1A	n.d.	n.d.	0.0463	n.d.	n.d.	0.0011	n.d.	n.d.	n.d.	n.d.
NN1B	n.d.	n.d.	0.1351	0.2138	0.0124	0.0035	n.d.	0.0002	n.d.	0.000006
NN1C	0.7714	n.d.	0.0562	n.d.	0.0099	0.0010	n.d.	0.0002	0.0003	n.d.
DM1A	1.7117	n.d.	0.0642	n.d.	n.d.	0.0016	0.0528	n.d.	0.0008	n.d.
DM1B	1.9464	n.d.	0.1150	0.0846	0.0082	0.0029	0.0501	0.0002	0.0009	0.000003
DM1C	2.0472	n.d.	0.0907	n.d.	0.0088	0.0019	0.0564	0.0002	0.0009	n.d.
DA1A	1.8317	n.d.	0.1012	n.d.	0.0167	0.0010	0.0645	0.0004	0.0008	n.d.
DA1B	1.4701	n.d.	0.1286	n.d.	n.d.	0.0012	0.0681	n.d.	0.0006	n.d.
DA1C	n.d.	n.d.	0.0558	n.d.	0.0200	0.0017	0.0800	0.0004	n.d.	n.d.
Phase 2 infant formulas										
ME2A	1.0969	n.d.	n.d.	n.d.	0.0138	n.d.	0.0393	0.0003	0.0005	n.d.
ME2B	1.2374	n.d.	n.d.	0.1003	0.0076	n.d.	0.0334	0.0001	0.0006	0.000003
ME2C	0.7569	n.d.	0.0304	n.d.	0.0090	0.0006	0.0264	0.0002	0.0003	n.d.
NC2A	n.d.	n.d.	0.1075	0.0461	0.0107	0.0018	0.0097	0.0002	n.d.	0.000001
NC2B	0.5848	n.d.	0.1392	n.d.	0.0165	0.0029	0.0208	0.0003	0.0002	n.d.
NC2C	0.6875	n.d.	0.1131	n.d.	0.0136	0.0022	0.0124	0.0003	0.0003	n.d.
NN2A	0.5212	n.d.	0.0363	n.d.	0.0062	0.0011	0.0063	0.0001	0.0003	n.d.
NN2B	0.4957	n.d.	0.0611	n.d.	0.0063	0.0018	0.0062	0.0001	0.0002	n.d.
NN2C	0.6598	n.d.	0.0392	n.d.	n.d.	0.0012	0.0050	n.d.	0.0003	n.d.
DM2A	1.8267	n.d.	0.0680	n.d.	0.0061	0.0011	0.0411	0.0001	0.0006	n.d.
DM2B	1.5085	n.d.	0.0566	n.d.	n.d.	0.0007	0.0516	n.d.	0.0005	n.d.
DM2C	1.5130	n.d.	0.0542	n.d.	n.d.	0.0006	0.0363	n.d.	0.0005	n.d.
DA2A	1.0955	n.d.	0.0679	n.d.	0.0195	0.0009	0.1013	0.0004	0.0003	n.d.
DA2B	1.3109	n.d.	0.0544	n.d.	0.0167	0.0008	0.0386	0.0003	0.0006	n.d.
DA2C	1.1523	n.d.	0.0646	n.d.	0.0198	0.0009	0.0981	0.0004	0.0005	n.d.

Different superscript letters in the same column indicate significant differences between the IFs at a significance level of *p* < 0.01. IFs, infant formulas; n.d., not determined. The THQs were calculated by the ratio between each ingested contaminant (infant formula ingestion rate) considering the sub-chronic reference oral dose (RfDo). THQ ≤ 1, indicates a negligible risk; THQ > 1, indicates potential harmful effects to infant health. TCR values between 10^−4^–10^−6^ indicate negligible evident cancer risk to infant health.

## Data Availability

Data that supports the findings of these experiments are available upon request.

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
