# Peer review of "Toxic Metals and Metalloids in Infant Formulas Marketed in Brazil, and Child Health Risks According to the Target Hazard Quotients and Target Cancer Risk"

_ijerph, 2022, doi:10.3390/ijerph191811178_

Round 1
Reviewer 1 Report
The manuscript presents us with a work that, although it is not new to determine potentially toxic metals or metalloids in food products for newborns, however, it is important to know and evaluate the risks and that was with the evaluation of the associated risks having been determined the Targets Hazard Quotients (THQ) and Target Cancer Risk (TCR).
The manuscript has a well-presented introduction, including the objectives of the work, the methods are adequate and well described, allowing them to be replicated by other authors. A good comparative statistical analysis of the results was carried out and the conclusions are in line with the results obtained.
I just have a note for the authors: Please make the footnote of table 5 fully visible in the manuscript.
Author Response
International Journal of Environmental Research and Public Health
Manuscript ID: ijerph-1876799
Title: Toxic metals and metalloids in infant formulas marketed in Brazil and potential child health risk assessment.
Authors: Cristine Couto Almeida, Diego dos Santos Baião, Paloma Almeida Rodrigues, Tatiana Dillenburg Saint’Pierre, Rachel Ann Hauser-Davis, Katia Christina Leandro, Vania Margaret Flosi Paschoalin, Marion Pereira Costa, Carlos Adam Conte-Junior
GENERAL COMMENTS BY THE AUTHORS
We believe that we have fully addressed all reviewers concerns and comments. Several modifications were carried out in the revised manuscript.
All modifications were highlighted in yellow.
Modifications suggested by the reviewers have polished the manuscript and increased its overall impact. We would like to thank reviewers for his/her insights and thoughtful critiques of our manuscript. By following the reviewer`s concerns, several points in the manuscript were better addressed and discussed, improving reader understanding.
After performing the modifications suggested by the reviewers, the entire text was revised by an editing specialized company in order to improve English grammar and syntax.
Review Report, Reviewer 1
Comments and Suggestions for Authors:
The manuscript presents us with a work that, although it is not new to determine potentially toxic metals or metalloids in food products for newborns, however, it is important to know and evaluate the risks and that was with the evaluation of the associated risks having been determined the Targets Hazard Quotients (THQ) and Target Cancer Risk (TCR).
The manuscript has a well-presented introduction, including the objectives of the work, the methods are adequate and well described, allowing them to be replicated by other authors. A good comparative statistical analysis of the results was carried out and the conclusions are in line with the results obtained.
I just have a note for the authors: Please make the footnote of table 5 fully visible in the manuscript.
Answer: We agree with the reviewer`s suggestion and we have improved the visibility of the footnote in Table 5 (page 10).

Reviewer 2 Report
The aim of the study was to determine the levels of toxic elements in infant food formulas available on the Brazilian market, recommended for infants aged 0 –6 months and 6 –12 months. The introduction is good but can be improved by stating the recommended tolerable limits of the elements investigated. Although reference is made to tolerable limits within the body of the article, these figures are not apparent to the reader until the supplementary data is viewed. The research is well-designed and the methodologies used are appropriate. However, the methodology does not describe how consumption data for the infant formula was collected. Did the authors use a 24- or 48-hour dietary recall to determine the frequency of consumption? How were the quantities estimated? Did they rely on intake rates based on published dietary survey data? This is an important part of the work and must be included. The authors also fail to indicate if the cancer slope factors used for the carcinogenic metals and metalloids have been adjusted for age. If this was done, it should be stated clearly in the methodology. If not, a clear justification must be provided with appropriate references. Parts of the conclusions are not based on the results provided. The authors did not provide the list of ingredients for the infant formulas sampled, but they attribute the high As levels to potentially contaminated grains used to fortify infant formula. This conclusion is not borne out by the results presented.
Suggested Edits
Line 175 – 176: The sentence is not complete, and the meaning is not clear. Are the authors defining TCR or explaining how they estimated it in this study?
Line 252 – 253: Makes reference to tolerable and safe levels of the metals and metalloids investigated but these are not present anywhere within the write-up.
Line 344-346: Requires editing to clarify the meaning of the sentence. The ‘they’ in “but they should be differentiated from those systems” is ambiguous. Does it refer to various organs or harmful outcomes?
Line 440-444: Kindly revise and split the sentence. It is too long and convoluted in its current state.
Author Response
International Journal of Environmental Research and Public Health
Manuscript ID: ijerph-1876799
Title: Toxic metals and metalloids in infant formulas marketed in Brazil and potential child health risk assessment.
Authors: Cristine Couto Almeida, Diego dos Santos Baião, Paloma Almeida Rodrigues, Tatiana Dillenburg Saint’Pierre, Rachel Ann Hauser-Davis, Katia Christina Leandro, Vania Margaret Flosi Paschoalin, Marion Pereira Costa, Carlos Adam Conte-Junior
GENERAL COMMENTS BY THE AUTHORS
We believe that we have fully addressed all reviewer concerns and comments. Several modifications were carried out in the revised manuscript.
All modifications were highlighted in yellow.
Modifications suggested by the reviewers have polished the manuscript and increased its overall impact. We would like to thank reviewers for his/her insights and thoughtful critiques of our manuscript. By following the reviewer`s concerns, several points in the manuscript were better addressed and discussed, improving reader understanding.
After performing the modifications suggested by the reviewers, the entire text was revised by an editing specialized company in order to improve English grammar and syntax.
Review Report (Reviewer 2)
Comments and Suggestions for Authors:
The aim of the study was to determine the levels of toxic elements in infant food formulas available on the Brazilian market, recommended for infants aged 0–6 months and 6–12 months. The introduction is good but can be improved by stating the recommended tolerable limits of the elements investigated. Although reference is made to tolerable limits within the body of the article, these figures are not apparent to the reader until the supplementary data is viewed. The research is well-designed and the methodologies used are appropriate. However, the methodology does not describe how consumption data for the infant formula was collected. Did the authors use a 24- or 48-hour dietary recall to determine the frequency of consumption? How were the quantities estimated? Did they rely on intake rates based on published dietary survey data? This is an important part of the work and must be included. The authors also fail to indicate if the cancer slope factors used for the carcinogenic metals and metalloids have been adjusted for age. If this was done, it should be stated clearly in the methodology. If not, a clear justification must be provided with appropriate references. Parts of the conclusions are not based on the results provided. The authors did not provide the list of ingredients for the infant formulas sampled, but they attribute the high As levels to potentially contaminated grains used to fortify infant formula. This conclusion is not borne out by the results presented.
Answer: We agree with the reviewer`s suggestion. In the present study data were not collected by 24- or 48-h dietary recall concerning infant formula consumption. This was not the purpose herein. Instead, the consumption recommendation stated on the infant formula labels was taken in consideration. The consumption frequency recommended for infant formulas was used to assess the toxicological risk by calculating the THQ for each toxic element present or not in infant formulas, always following label ingestion recommendations. This information was added in section 2.4.1. Target hazard quotient (page 4, lines 157-162).
The oral cancer slope factors (CFSo) are considered to estimate cancer risk, and the reference oral doses (RfDo) are used to calculate the hazard quotient. Both values refer to the general population, including sensitive subgroups. In this study, in order to con-sider the age group evaluated, the average body weights for each age group (0-6 months and 7-12 months) were considered for the risk assessment calculations [U.S. EPA, 2009; U.S. EPA, 2022]. Dose-response assessments regarding carcinogen risk define the relationship be-tween the dose of a toxic agent and its putative carcinogenic effect if it is classified as a carcinogenic for human beings [Nurchi et al., 2012]. Data obtained in dose-response assessments are combined with exposure assessment data to estimate risks. When the risks to short expo-sure or prolonged exposures cannot be measured using animal models or data from epidemiologic studies, mathematical models should be applied. When data are limited and when the mechanisms of carcinogenic action are uncertain, linear models or procedures are adopted if compatible with the available information. The U.S. Environ-mental Protection Agency (U.S. EPA) usually employs the linearized multistage procedure in the absence of adequate information (https://www.epa.gov/iris/epas-approach-assessing-risks-associated-chronic-exposure-carcinogens). These information were added in the manuscript discussion (page 9, lines 323-338).
The application of slope factors and unit risks from U.S. EPA for foodstuff, considers the multiplication of the slope factor (risk per mg/kg/day), the concentration of the chemical element in the evaluated foodstuff (ppm) and the daily intake (in mg) of the food (U.S. EPA, 2009). Therefore, the total dietary risk is found by summing risks across all foods. The only difference from the THQ to the TCR is that oral reference dose (RfDo), which is used for the determination of THQ, and an oral slope factor (CSFO) that is utilized by TCR. This CSFO is usually expressed in units of proportion (of a population) affected per mg of substance/kg body weight-day and is generally reserved for use in the low-dose region of the dose-response relationship, that is, for exposures corresponding to risks less than 1 in 100 [Antoine et al., 2017].
Finally, the conclusion was modified by including novel Information that better corroborates the results of the present study (pages 12, lines 438-457).
Reference:
Antoine, J., Fung, L., & Grant, C. N. (2017). Assessment of the potential health risks associated with the aluminium, arsenic, cadmium and lead content in selected fruits and vegetables grown in Jamaica. Toxicology reports, 4, 181–187. https://doi.org/10.1016/j.toxrep.2017.03.006
Nurchi, V. M., Crisponi, G., Bertolasi, V., Faa, G., Remelli, M. (2012). Aluminium-dependent human diseases and chelating properties of aluminium chelators for biomedical applications. Metal Ions in Neurological Systems, 10, 103-123. doi: 10.1007/978-3-7091-1001-0_10
United States Environmental Protection Agency (U.S. EPA). (2009). Risk-assessment guidance for superfund. Volume 1. Human health evaluation manual. Part A. Interim report (final). Retrieved from https://www.epa.gov/sites/default/files/2015-09/documents/partf_200901_final.pdf Accessed November 11, 2021.
United States Environmental Protection Agency (U.S. EPA). (2022). Regional Screening Level (RSL) Subchronic Toxicity Supporting. Retrieved from https://semspub.epa.gov/work/HQ/402403.pdf Accessed July 04, 2022
Suggested Edits
Line 175-176: The sentence is not complete, and the meaning is not clear. Are the authors defining TCR or explaining how they estimated it in this study?
Answer: We agree with the reviewer’s correction and the sentence was reformulated (page 4, lines 180-182). The target cancer risk (TCR) was used to assess the potential risk associated with exposure to carcinogens during the lifetime exposure period. In the present study, Al, iAs and Pb were considered potentially carcinogenic, so the TCR was calculated only for these elements.
Line 252-253: Makes reference to tolerable and safe levels of the metals and metalloids investigated but these are not present anywhere within the write-up.
Answer: We agree with the reviewer`s suggestion and indicate that the tolerable and safe levels for metal and metalloids were addressed in the supplemental file Table S3 (page 6, line 254-257).
Line 344-346: Requires editing to clarify the meaning of the sentence. The ‘they’ in “but they should be differentiated from those systems” is ambiguous. Does it refer to various organs or harmful outcomes?
Answer: We agree with the reviewer’s correction and the final part of the sentence was removed (page 10, lines 352-354).
Line 440-444: Kindly revise and split the sentence. It is too long and convoluted in its current state.
Answer: We agree with the reviewer’s suggestion and the conclusion section was reformulated (page 12, lines 438-457).
Reviewer 3 Report
Dear editor,
The submitted manuscript describes an analysis of a sample of infant formula powder in Brazil for metals and metalloids, and a deterministic, single-point health risk assessment due to exposure to toxigenic and carcinogenic compounds in the analyzed formula samples. Although the metal contamination in infant formula is an important threat that needs to be studied extensively, I would like to point out a few important points in terms of risk assessment and communication.
Although the deterministic approach is commonly used in toxicology risk assessments, it has considerable shortcomings that need to be addressed. A deterministic approach will not inform the risk managers (which is the ultimate purpose of a risk assessment) accurately, as a single-point estimate is far from describing the total population. Risk is inherently a probabilistic measure, therefore, the analysis should involve probability. In the current analysis, it is also impossible to assess which variables contribute the most to the risk (Sensitivity analysis); therefore, interventions or increased surveillance can be addressed. I suggest revisiting the analysis form a probabilistic point of view (Monte-Carlo simulation), if the time permits. If not possible, a few different scenarios should be included in the analysis, i.e. worst case, best case and most likely. Ultimately, these estimates will aid in setting science-based limits for toxic metal and metalloid exposure from consumption of infant formula. At least, a narrative discussion should be included that the risk estimates are deterministic.
Wording through the manuscript should be revised carefully. There is no such thing as “no risk” or “zero risk”. Risk can be defined as a probability distribution that is never zero. There is always variability and uncertainty. As the US EPA suggests, deterministic ratios to describe risks are only “screening level estimates” which indicate high or low risk situations. There are a lot of variable situations which would change the risk estimates significantly. Brazil is a very large country, therefore it is inaccurate to consider all infants are exposed to same quantities and their metabolism are exactly the same with each other.
In the present study, authors collected 30 samples from grocery stores around Brazil. However, it was mentioned through the manuscript that the results here would represent all the infant formula sold in Brazil. Thirty samples are far from enough to make a conclusion about the whole industry and a lot of variability and uncertainty is expected around these estimates. It is wrong to conclude that the Brazilian infant formulas were “free of Hg” or “low contents of Cb, Pb and U”. I challenge that another 30 samples would yield very different results. Therefore, the authors should refrain from using a language that implies an adequate number of samples were taken that represent the whole country.
A thorough grammar and language check should be conducted by a native speaker before resubmission.
Abstract
Line 34: Contamination or exposure?
Line 39: Aluminum “ranging from”
Line 41: Lowest concentrations? LOD? LOQ?
Lines 43-45L There is no such thing as “no risk”, zero risk cannot exist. You can say negligible or minimal, but the risk is still a statistical distribution. You don’t observe risk, you estimate it.
Introduction
Line 51: are->is
Lines 70-71: Also in concentrated liquid form that water is added.
Line 95: stistly?
Materials and Methods
Are the given RfDo values for infants, adults or general population?
“no health risk”. In how many of the exposure events, will that ratio be below 1? It is a variable and uncertain event; therefore, I am sure there will be some extremities, so you should not say no risk, but it is unlikely.
What would be the cumulative hazard for continuous exposure to several chemicals, like a hazard index?
Line 151: Risk of what?
Line 193: Why repeated measures? You don’t have time as a variable in the model. Were the samples taken at different times after a treatment?
Results & Discussion
Line 202: Please first define the abbreviation and then use.
Lines 213-220: This paragraph is not results or discussion. Should be moved to introduction.
Lines 221-224: Same.
Table 3: Please also indicate the number of samples each mean and SD originate from. Consider providing averages for each element tested across different brands.
Table 3 and 4 are essentially the same thing.
Line 226: What variability? How did you quantify this variability?
Line 262: Is this the result of a statistical analysis? Please give p-values if so.
Lines 280-283: Were these “considerable variations” included in the risk model?
Lines 280-315: I did not understand the relevance of these three paragraphs to the results or discussion from this study. General information can be given in the appropriate section, but this section should be within the scope of your study.
Table 5: Counts below LOQ and LOD can be considered as “censored values” and there are ways to estimate values from these ranges to be used in risk assessments. Have you tried any of these methods, how would this change your conclusions about the risks?
Line 358: Exposure?
Lines 362-365: “risk of exposure” is a confusing term. Are you calculating “risk of exposure” or the risk of adverse health effects? These need to be clarified. Following a formal risk assessment framework (i.e., hazard identification, exposure assessment, risk characterization) will make your work a lot easier to differentiate between the hazard, the exposure, and the risks.
Lines 367-369: What is this safe range?
Line 387: Exposure
Line 414: What are the units for these numbers? “Levels” don’t mean anything without a unit.
Line 428: How can you say “free of Hg”, when you have a detection limit?
Lines 440-448: Authors suggest increased inspection for toxic metal and metalloids contamination in the products. A complete risk assessment should also guide the regulatory and industry risk managers to decide on the limits that should be established or updated based on the results from the risk assessment. I believe these recommendations are essential to a risk study and should be addressed in the current manuscript.
Tables
Please follow the rules for significant figures in tables and any reported numerical results.
Author Response
International Journal of Environmental Research and Public Health
Manuscript ID: ijerph-1876799
Title: Toxic metals and metalloids in infant formulas marketed in Brazil and potential child health risk assessment.
Authors: Cristine Couto Almeida, Diego dos Santos Baião, Paloma Almeida Rodrigues, Tatiana Dillenburg Saint’Pierre, Rachel Ann Hauser-Davis, Katia Christina Leandro, Vania Margaret Flosi Paschoalin, Marion Pereira Costa, Carlos Adam Conte-Junior
GENERAL COMMENTS BY THE AUTHORS
We believe that we have fully addressed all reviewer concerns and comments. Several modifications were carried out in the revised manuscript.
All modifications were highlighted in yellow.
Modifications suggested by the reviewers have polished the manuscript and increased its overall impact. We would like to thank reviewers for his/her insights and thoughtful critiques of our manuscript. By following the reviewer`s concerns, several points in the manuscript were better addressed and discussed, improving reader understanding.
After performing the modifications suggested by the reviewers, the entire text was revised by an editing specialized company in order to improve English grammar and syntax.
Review Report (Reviewer 3)
Comments and Suggestions for Authors:
The submitted manuscript describes an analysis of a sample of infant formula powder in Brazil for metals and metalloids, and a deterministic, single-point health risk assessment due to exposure to toxigenic and carcinogenic compounds in the analyzed formula samples. Although the metal contamination in infant formula is an important threat that needs to be studied extensively, I would like to point out a few important points in terms of risk assessment and communication.
Although the deterministic approach is commonly used in toxicology risk assessments, it has considerable shortcomings that need to be addressed. A deterministic approach will not inform the risk managers (which is the ultimate purpose of a risk assessment) accurately, as a single-point estimate is far from describing the total population. Risk is inherently a probabilistic measure, therefore, the analysis should involve probability. In the current analysis, it is also impossible to assess which variables contribute the most to the risk (Sensitivity analysis); therefore, interventions or increased surveillance can be addressed. I suggest revisiting the analysis form a probabilistic point of view (Monte-Carlo simulation), if the time permits. If not possible, a few different scenarios should be included in the analysis, i.e. worst case, best case and most likely. Ultimately, these estimates will aid in setting science-based limits for toxic metal and metalloid exposure from consumption of infant formula. At least, a narrative discussion should be included that the risk estimates are deterministic.
Answer: We disagree with the reviewer’s suggestion. The aim of the present study was to evaluate the presence or absence of toxic metals in infant formulas. The present study did not aim to assess risk managers or other risk variables that may have contributed to metal contamination. For this, we would need to consider variables of the entire production chain of infant formula for each brand or product. Therefore, we raised the possible causes or events that may have led to the increase in toxic metals identified in the analyzed products in the discussion topic.
Wording through the manuscript should be revised carefully. There is no such thing as “no risk” or “zero risk”. Risk can be defined as a probability distribution that is never zero. There is always variability and uncertainty. As the US EPA suggests, deterministic ratios to describe risks are only “screening level estimates” which indicate high or low risk situations. There are a lot of variable situations which would change the risk estimates significantly. Brazil is a very large country, therefore it is inaccurate to consider all infants are exposed to same quantities and their metabolism are exactly the same with each other.
Answer: We agree with the reviewer’s correction. The terms “zero risk” or “no risk” were replaced by the terms “negligible risk” or “minimal risk” throughout the manuscript.
In the present study, authors collected 30 samples from grocery stores around Brazil. However, it was mentioned through the manuscript that the results here would represent all the infant formula sold in Brazil. Thirty samples are far from enough to make a conclusion about the whole industry and a lot of variability and uncertainty is expected around these estimates. It is wrong to conclude that the Brazilian infant formulas were “free of Hg” or “low contents of Cb, Pb and U”. I challenge that another 30 samples would yield very different results. Therefore, the authors should refrain from using a language that implies an adequate number of samples were taken that represent the whole country.
Answer: We agree with the reviewer’s correction and all sentence aforementioned were modified to improve understanding of manuscript.
A thorough grammar and language check should be conducted by a native speaker before resubmission.
Answer: We agree with the reviewer’s suggestion, and the revised manuscript was entirely revised by a native speaker, in order to improve grammar and syntax.
Abstract
Line 34: Contamination or exposure?
Answer: We agree with the reviewer’s correction and the word contamination was replaced by exposure (page 1, line 34).
Line 39: Aluminum “ranging from”
Answer: We agree with the reviewer’s correction and the sentence was corrected as suggested (page 1, line 39).
Line 41: Lowest concentrations? LOD? LOQ?
Answer: The correct words are the lowest concentrations. We have revised this (page 1, lines 41-42).
Lines 43-45L There is no such thing as “no risk”, zero risk cannot exist. You can say negligible or minimal, but the risk is still a statistical distribution. You don’t observe risk, you estimate it.
Answer: We agree with the reviewer’s correction and the terms “zero risk” or “no risk” were replaced by the terms “negligible risk” or “minimal risk” in all manuscript, as suggested.
Introduction
Line 51: are->is
Answer: Modified, as suggested (page 2, line 51).
Lines 70-71: Also in concentrated liquid form that water is added.
Answer: We agree with the reviewer’s correction and we have added this information in the sentence as suggested (page 2, lines 71-72).
Line 95: stistly?
Answer: This word was corrected, as suggested (page 2, line 96).
Materials and Methods
Are the given RfDo values for infants, adults or general population?
Answer: The RfDo values are for the general population. This information was included in the revised manuscript (page 4, lines 157-162).
“no health risk”. In how many of the exposure events, will that ratio be below 1? It is a variable and uncertain event; therefore, I am sure there will be some extremities, so you should not say no risk, but it is unlikely.
Answer: We agree with the reviewer’s correction and the terms “no health risk” was replaced by the term “unlikely health risk” in the entire manuscript.
What would be the cumulative hazard for continuous exposure to several chemicals, like a hazard index?
Answer: The cumulative danger is related to those chemical contaminants that tend to accumulate in the body, which means they are not easily excreted. Therefore, the Hazard index, a widely used method to assess the cumulative risk and danger to humans after exposure to various food contaminants, was used based on the standard approaches described in the Guidelines for the Assessment of Health Risks of Chemical Mixtures of the United States Environmental Protection Agency (U.S. EPA). This was explained in material methods (page 4, line 176-178) and results section (page 9, lines 316-319).
Line 151: Risk of what?
Answer: Modified as suggested (page 4, line 155).
Line 193: Why repeated measures? You don’t have time as a variable in the model. Were the samples taken at different times after a treatment?
Answer: In statistics, the means of two or more samples can be compared in relation to some variable of interest. Then, tests were used to determine whether or not there are significant differences between the means of an individual sample. Therefore, it is necessary to perform the analyses on the same sample at least twice to obtain an average. In addition, the test used was Analysis of Variance or ANOVA, which is a procedure used to compare the distribution of three or more groups in independent samples. Repeated measures term refers to analyses that were performed in quintuplicate on each infant formula sample.
Results & Discussion
Line 202: Please first define the abbreviation and then use.
Answer: The abbreviations LOD and LOQ were described in material and methods section (page 4, line 149) and where then used throughout the entire manuscript.
Lines 213-220: This paragraph is not results or discussion. Should be moved to introduction.
Answer: We agree with the reviewer’s suggestion and this information was removed from the discussion.
Table 3: Please also indicate the number of samples each mean and SD originate from. Consider providing averages for each element tested across different brands.
Answer: Three batches of each infant formulas brand marketed in Brazil were analyzed in quintuplicate. So, about 150 analyzes were made. Therefore, the data were reported as means ± SD (N°: 150). This sentence was modified in Table 3 to improve the understanding (page 6).
Table 3 and 4 are essentially the same thing.
Answer: Table 3 refers to the average concentration of the batches of each infant formula, while table 4 refers to the concentration of the elements analyzed in each batch of infant formula. Both tables were included because the average from three batches could not reflect individual content of metals or metalloids in each brand.
Line 226: What variability? How did you quantify this variability?
Answer: Variability refers to the different evaluated lots, i.e., differences observed for Al results in infant formulas (shown in table 3). This sentence was modified to improve understanding (page 6, lines 224-229).
Line 262: Is this the result of a statistical analysis? Please give p-values if so.
Answer: We agree with the reviewer’s correction and the p-value was added in the manuscript (page 7, line 266).
Lines 280-283: Were these “considerable variations” included in the risk model?
Answer: No, these considerable variations were not included in the risk model because the aim of this sentence was just to report or discuss the greater susceptibility at this stage of life, which occurs due to the immaturity of the gastrointestinal system as referred in the literature. The sentence was modified to clarify this point (page 8, lines 285-294).
Lines 280-315: I did not understand the relevance of these three paragraphs to the results or discussion from this study. General information can be given in the appropriate section, but this section should be within the scope of your study.
Answer: We agree with the reviewer’s suggestion. The aim of these three paragraphs was to extensively discuss the risk of toxic metal and metalloid ingestion during childhood. These paragraphs also discuss the reasons that make children and infants more vulnerable to the ingestion of these metals and the mechanisms of action of the metals that were found in greater amounts of the infant formula analyzed in the present study.
Therefore, following the reviewer's recommendation, we have reduced this part of the discussion, covering only the essential information about the risks of consumption of these metals by infants (page 8, line 295-306).
Table 5: Counts below LOQ and LOD can be considered as “censored values” and there are ways to estimate values from these ranges to be used in risk assessments. Have you tried any of these methods, how would this change your conclusions about the risks?
Answer: No, this approach is not routinely employed in this type of analysis, as they may lead to biased results. Even if it were possible to measure these values and apply them in the calculations used in the risk assessment, we believe that the conclusion would still be a low probability of risk concerning the evaluated elements. Therefore, several works published by our research group do not estimate values below the LOQ and LOD to be used in risk assessments (Hauser-Davis et al., 2021; Almeida et al., 2022; Souza-Araujo et al., 2022; Rodrigues et al., 2022).
References:
- Alemida, C.C., Baião, D.S., Rodrigues, P.A., Saintpierre, T.D., Hauser-Davis, R.A., Leandro, K.C., Paschoalin, V.M.F., Costa, M.P. Conte-Junior, C.A. (2022). Macrominerals and Trace Minerals in Commercial Infant Formulas Marketed in Brazil: Compliance With Established Minimum and Maximum Requirements, Label Statements, and Estimated Daily Intake. Frontiers in nutrition, 9, 857-698. 10.3389/fnut.2022.857698
- Souza-Araujo, J., Hussey, N.E., Hauser-Davis, R.A., Rosa, A.H., Lima, M.O., Giarrizzo, T. (2022). Human risk assessment of toxic elements (As, Cd, Hg, Pb) in marine fish from the Amazon. Chemosphere, 301, 134-575. https://doi.org/10.1016/j.chemosphere.2022.134575
- Rodrigues, P.A., Ferrari, R.G., Rosario, D.K.A., Hauser-Davis, R.A., Conte-Junior, C.A. (2022). Mercurial Contamination: A Consumer Health Risk Assessment Concerning Seafood from a eutrophic estuary in Southeastern Brazil. Frontiers in Marine Science, 9, 765323. https://doi.org/10.3389/fmars.2022.765323
- Hauser-Davis, R.A., Rocha, R.C.C., Saintpierre, T.D., Adams, D.H. (2021). Metal concentrations and metallothionein metal detoxification in blue sharks, Prionace glauca from the Western North Atlantic Ocean. Journal of Trace Elements in Medicine and Biology, 68, 126813. https://doi.org/10.1016/j.jtemb.2021.126813
Line 358: Exposure?
Answer: Modified, as suggested (page 11, line 365).
Lines 362-365: “risk of exposure” is a confusing term. Are you calculating “risk of exposure” or the risk of adverse health effects? These need to be clarified. Following a formal risk assessment framework (i.e., hazard identification, exposure assessment, risk characterization) will make your work a lot easier to differentiate between the hazard, the exposure, and the risks.
Answer: We agree with the reviewer’s correction and we have altered the term “risk of exposure” to “risk of adverse health effects” (page 11, lines 372-373).
Lines 367-369: What is this safe range?
Answer: We agree with the reviewer’s correction and this sentence was modified (page 11, lines 375-378).
Line 387: Exposure
Answer: Modified as suggested (page 11, line 396).
Line 414: What are the units for these numbers? “Levels” don’t mean anything without a unit.
Answer: We agree with the reviewer’s suggestion and the units for each result were added to the revised manuscript (page 12, lines 423-424).
Line 428: How can you say “free of Hg”, when you have a detection limit?
Answer: We agree with the reviewer’s suggestion and the sentence was reformulated (page 12, line 438).
Lines 440-448: Authors suggest increased inspection for toxic metal and metalloids contamination in the products. A complete risk assessment should also guide the regulatory and industry risk managers to decide on the limits that should be established or updated based on the results from the risk assessment. I believe these recommendations are essential to a risk study and should be addressed in the current manuscript.
Answer: We agree with the reviewer’s suggestion and the sentence was rewritten (page 12, lines 447-458).
Tables
Please follow the rules for significant figures in tables and any reported numerical results.
Answer: The choice for the number of decimal places used respected the presentation of the smallest number, that is, we standardized the presentation of the results in the tables with 4 decimal places.

Round 2
Reviewer 3 Report
Thanks to the authors for the providing the revision.
For my first comment in R1, I don't think I was clear enough, so I wanted to clarify a little bit more. As mentioned in the Introduction section, this study clearly aims to "assess health risks" and the aim of a risk assessment is to inform risk management and risk communication activities. However, the current manuscript only reports risk as a single number. Does this number represent a best-case or worst-case scenario? How could this number change if the contamination levels were at the upper or lower confidence intervals? The answers to questions like these would help the regulatory and industry risk managers to set risk-based targets.
I just think the message would be clearer if the uncertainty around these estimates would be explained. This can be done by a stochastic process, however it is also possible to discuss these qualitatively, so that the reader is aware of the shortcomings of the proposed risk assessment outcomes.
Author Response
International Journal of Environmental Research and Public Health
Manuscript ID: ijerph-1876799
Title: Toxic metals and metalloids in infant formulas marketed in Brazil and child health risk assessment.
Authors: Cristine Couto Almeida, Diego dos Santos Baião, Paloma Almeida Rodrigues, Tatiana Dillenburg Saint’Pierre, Rachel Ann Hauser-Davis, Katia Christina Leandro, Vania Margaret Flosi Paschoalin, Marion Pereira Costa, Carlos Adam Conte-Junior
GENERAL COMMENTS BY THE AUTHORS
We believe that we have fully addressed all reviewer concerns and comments. The modifications were carried out in the revised manuscript. All modifications were highlighted in yellow.
Modifications suggested by the reviewer have polished the manuscript and increased its overall impact. We would like to thank reviewer for his/her insights and thoughtful critiques of our manuscript. By following the reviewer concerns, points in the manuscript were better addressed and discussed, improving reader understanding.
Review 3 Report
Comments and Suggestions for Authors
Thanks to the authors for the providing the revision.
For my first comment in R1, I don't think I was clear enough, so I wanted to clarify a little bit more. As mentioned in the Introduction section, this study clearly aims to "assess health risks" and the aim of a risk assessment is to inform risk management and risk communication activities. However, the current manuscript only reports risk as a single number. Does this number represent a best-case or worst-case scenario? How could this number change if the contamination levels were at the upper or lower confidence intervals? The answers to questions like these would help the regulatory and industry risk managers to set risk-based targets.
I just think the message would be clearer if the uncertainty around these estimates would be explained. This can be done by a stochastic process, however it is also possible to discuss these qualitatively, so that the reader is aware of the shortcomings of the proposed risk assessment outcomes.
Answer: As mentioned by the reviewer, one of the objectives of the study was to assess the likely health risk arising from the ingestion of toxic metals and metalloids through the consumption of infant formula. In our study, this risk was evaluated through the identification and quantification of the toxic metals and metalloids presents in infant formulas. Based on these results, we used these values ​​in the equation that evaluates the toxicological risk of children in toxic metals and non-carcinogenic metalloids (THQ) and in the equation that assesses the potential risk associated with the exposure of carcinogenic toxic metals and metalloids (TCR) identified in infant formulas. The calculated THQ and TCR values that were based on the average of the values ​​detected for each toxic metal or metalloid in infant formulas. Therefore, the values ​​reported in the present study for THQ and TCR did not address the best or worst scenario, but rather the average of the toxic metals determined in the assessed infant formulas.
Most of the studies published in the literature performed estimations of these indices in exactly the same way as we approached in the present manuscript (please see Eticha et al., 2018; Su et al., 2020; BaÅŸaran, 2022; Rodrigues et al., 2022; Souza-Araujo et al., 2022). Furthermore, as the reviewer him/herself stated at the end of the questioning, “it is possible to discuss the results qualitatively, however, leaving a clear message regarding how the calculation was performed so that the reader is aware of the deficiencies in the results of the proposed risk assessment. Therefore, this information was introduced in the discussion section (page 9, line 338-341) for a better understanding of the reported estimates.
In general, we agree that the use of the expression "risk assessment" in this work was used incorrectly. This is because the concept of “risk assessment” is an organized process used to describe and estimate the probability of adverse health outcomes from environmental exposures to chemicals using four steps such as hazard identification, dose-response assessment, exposure assessment and risk characterization (U.S. EPA, 2009).
Therefore, the term risk assessment has been replaced throughout the manuscript and in the title, after considering valuable concerns of the reviewer.
Cited references:
- BaÅŸaran, B. (2022). An assessment of heavy metal level in infant formula on the market in Turkey and the hazard index. Journal of Food Composition and Analysis, 105, 1-9. https://doi.org/10.1016/j.jfca.2021.104258
- Eticha, T., Afrasa, M., Kahsay, G., Gebretsadik, H. (2018). Infant Exposure to Metals through Consumption of Formula Feeding in Mekelle, Ethiopia. International Journal of Analytical Chemistry, 2018(2985698), 1-5. https://doi.org/10.1155/2018/2985698
- Rodrigues, P.A., Ferrari, R.G., Rosario, D.K.A., Hauser-Davis, R.A., Conte-Junior, C.A. (2022). Mercurial Contamination: A Consumer Health Risk Assessment Concerning Seafood from a eutrophic estuary in Southeastern Brazil. Frontiers in Marine Science, 9, 765323. https://doi.org/10.3389/fmars.2022.765323
- Souza-Araujo, J., Hussey, N.E., Hauser-Davis, R.A., Rosa, A.H., Lima, M.O., Giarrizzo, T. (2022). Human risk assessment of toxic elements (As, Cd, Hg, Pb) in marine fish from the Amazon. Chemosphere, 301, 134-575. https://doi.org/10.1016/j.chemosphere.2022.134575
- Su, C., Zheng, N., Gao, Y., Huang, S., Yang, X., Wang, Z., Yang, H., Wang, J. (2020). Content and Dietary Exposure Assessment of Toxic Elements in Infant Formulas from the Chinese Market. Foods, 9(12), 1-10. https://doi.org/10.3390/foods9121839
- United States Environmental Protection Agency (U.S. EPA). (2009). Risk-assessment guidance for superfund. Volume 1. Human health evaluation manual. Part A. Interim report (final). Retrieved from https://www.epa.gov/sites/default/files/2015-09/documents/partf_200901_final.pdf
